# Effect of α-tubulin acetylation on the doublet microtubule structure

**Shun Kai Yang[1†‡], Shintaroh Kubo[1†], Corbin Steven Black[1], Katya Peri[1], Daniel Dai[1], Thibault Legal[1], Melissa Valente-Paterno[1], Jacek Gaertig[2], Khanh Huy Bui[1,3]\***

[1]Department of Anatomy and Cell Biology, McGill University, Montréal, Canada;
[2]Department of Cellular Biology, University of Georgia, Athens, United States;
[3]Centre de Recherche en Biologie Structurale, McGill University, Montréal, Canada

**Abstract** Acetylation of α-tubulin at the lysine 40 residue (αK40) by αTAT1/MEC-17 acetyltransferase modulates microtubule properties and occurs in most eukaryotic cells. Previous literatures suggest that acetylated microtubules are more stable and damage resistant. αK40 acetylation is the only known microtubule luminal post-translational modification site. The luminal location suggests that the modification tunes the lateral interaction of protofilaments inside the microtubule. In this study, we examined the effect of tubulin acetylation on the doublet microtubule (DMT) in the cilia of *Tetrahymena thermophila* using a combination of cryo-electron microscopy, molecular dynamics, and mass spectrometry. We found that αK40 acetylation exerts a small-scale effect on the DMT structure and stability by influencing the lateral rotational angle. In addition, comparative mass spectrometry revealed a link between αK40 acetylation and phosphorylation in cilia.

## eLife assessment

This **fundamental** study employs a combination of cryo-electron microscopy, molecular dynamics, and mass spectrometry to elucidate the role of α-tubulin acetylation at the lumenal lysine 40 residue (αK40) within the cilium. **Compelling** evidence shows αK40 acetylation to impact the structure and stability of doublet microtubules in cilia by affecting the lateral rotational angle. The work will be of relevance to those interested in cytoskeleton and structural biology.

## Introduction

Cilia have diverse roles in cell motility, sensory functions, signaling, and growth control. Motile cilia drive the flow of fluid, including human sperm (*Lehti and Sironen, 2017*), mucus clearance in the respiratory tract (*Bustamante-Marin and Ostrowski, 2017*), and cerebrospinal fluid circulation (*Djenoune and Wyart, 2017*). Non-motile primary cilia in rod cells are critical for transmitting chemical signals converted from light (*Sjostrand, 1953*). In kidney epithelial cells, cilia function as mechanosensors for transmitting fluid flow signals into signaling pathways. At the core, motile and primary cilia share the same cytoskeletal framework, the axoneme (*Porter and Sale, 2000*), composed of a bundle of nine outer doublet microtubules (DMTs). Each DMT is composed of protofilaments (PFs) of tubulins that form into a hollow cylinder A-tubule and an incomplete cylinder B-tubule (*Figure 1A*). Inside the DMT lumen, a weaving network of microtubule inner proteins (MIPs) stabilizes the DMT structure (*Ichikawa et al., 2019*; *Ichikawa et al., 2017*; *Ma et al., 2019*).

Tubulins undergo several highly conserved post-translational modifications (PTMs) that collectively represent the so-called 'tubulin code', in which PTMs modulate microtubule properties directly or indirectly through the binding of microtubule-associated proteins. The common and well-studied tubulin PTMs are phosphorylation, detyrosination, glutamylation, glycylation, and acetylation (*Wloga*

**\*For correspondence:**
huy.bui@mcgill.ca

[†]These authors contributed equally to this work

**Present address:** [‡]Department of Biochemistry, University of Oxford, Oxford, United Kingdom

**Competing interest:** The authors declare that no competing interests exist.

*et al., 2017*). DMT contains a unique signature of PTMs including cilia-specific PTMs such as glycylation. For example, the B-tubule of DMT is enriched with glutamylated (*Lechtreck and Geimer, 2000*) and detyrosinated (*Johnson, 1998*) tubulins, while A-tubule tubulins are mostly unmodified. PTM is a fine tune for ciliary function rather than a biphasic switch. Glutamylation could regulate inner dynein arm activities, which control the ciliary waveform (*Kubo et al., 2010*; *Lechtreck and Geimer, 2000*; *Suryavanshi et al., 2010*). Hyperglutamylation due to depletion of deglutamylases can improve intraflagellar transport in ift88-deficient zebrafish (*Pathak et al., 2014*). Lack of glycylation causes abnormal pre- and post-powerstroke conformations of dynein arms in mouse sperm (*Gadadhar et al., 2021*). The loss of glycylation sometimes leads to an increase in glutamylation, possibly because the two PTMs compete for the same set of modification sites (glutamic acids) on tubulins (*Kubo et al., 2015*; *Rogowski et al., 2009*; *Wloga et al., 2009*). These observations suggest that there is an interplay between PTMs and cilia properties.

One of the most intriguing PTMs in cilia is acetylation, which occurs inside the lumen of DMT on the lysine 40 residue of α-tubulins (αK40). αK40 acetylation in *Chlamydomonas reinhardtii* cilia was the first identified tubulin acetylation (*LeDizet and Piperno, 1987*; *L'Hernault and Rosenbaum, 1983*). Tubulin acetylation was later found on different microtubules in cells, such as in neurons (*Fukushige et al., 1999*). Acetylation could also take place on lysine 60 and lysine 370 residues of α-tubulin and lysine 58 of β-tubulin (*Liu et al., 2015*). Acetylation is interesting because the αK40 loop is the only luminal PTM site and is close to the tubulin lateral interaction interface (*Kaul et al., 2014*; *Figure 1B*). The αK40 loop is flexible and close to the tubulin lateral interaction interface (*Eshun-Wilson et al., 2019*; *Howes et al., 2014*) and is almost 100% completely acetylated in cilia (*Akella et al., 2010*). Acetylated αK40 has been shown to enhance microtubule stability and longevity in vitro (*Schaedel et al., 2015*), while deacetylated microtubules decrease in rigidity and are prone to complete breakage events (*Xu et al., 2017*). Biochemical works on reconstituted microtubules from acetylated and deacetylated tubulins suggest that αK40 acetylation directly weakens inter-PF interactions and, hence enhance flexibility and reduce mechanical fatigue of microtubule (*Portran et al., 2017*).

The main enzyme responsible for αK40 acetylation is alpha-acetyltransferase-1 (αTAT1), also known as MEC-17 (*Akella et al., 2010*). Deacetylation is carried out by histone deacetylase 6 (HDAC6) (*Hubbert et al., 2002*) and nicotinamide adenine dinucleotide-dependent deacetylase sirtuin 2 (SIRT2) (*North et al., 2003*). Mice that lack αTAT1 have defective sperm flagellar beating (*Kalebic et al., 2013*), and *C. elegans* become touch insensitive without αTAT1 (*Akella et al., 2010*; *Shida et al., 2010*). Motor proteins travel preferentially on acetyl-K40 microtubules because of their higher binding affinity (*Garnham and Roll-Mecak, 2012*; *Reed et al., 2006*). Overexpression of HDAC6 or SIRT2 could lead to short cilia (*Ran et al., 2015*; *Zhou et al., 2014*), but the cilia-shortening effect of HDAC6 could be countered by an acetyl-K mimic α-tubulin, suggesting that it may destabilize the microtubule lattice (*Cueva et al., 2012*).

The αK40 loop is functionally important, but it is flexible and disordered even in both acetylated and deacetylated reconstituted microtubules (*Eshun-Wilson et al., 2019*; *Howes et al., 2014*). Cryo-EM and molecular dynamic studies of acetylated microtubules suggest that acetylation restricts the αK40 loop motion by disturbing the electrostatic interaction (*Eshun-Wilson et al., 2019*). This produces more ordered loops and stabilizes the microtubule lattice. In certain organisms, such as *C. elegans*, acetylation of αK40 would disrupt an intramonomer salt bridge from αE55 to αK40 (*Cueva et al., 2012*). Replacing αK40 with arginine or in the absence of acetyl-αK40 could result in the formation of a salt bridge from αE55 with αK40, αR40, or αH283, to change the inter-PF angle leading to elliptical microtubules and variation in the number of PFs (*Cueva et al., 2012*).

Interestingly, many αK40 loops are fully structured in the cryo-EM map of the DMT from ciliate and green algae (*Khalifa et al., 2020*; *Ma et al., 2019*). This result suggests that the αK40 loop may have an important role in recognizing and binding with different MIPs to stabilize cilia, and disruption of acetylation may disrupt the interaction between the αK40 loop and MIPs.

In this work, we aimed to clarify the role of αK40 acetylation in the assembly and stability of ciliary DMT. To answer our questions, we studied the structural effect of acetylated and non-acetylated tubulins on DMT and MIPs from *wild-type* (*WT*) *Tetrahymena thermophila*, mutants lacking tubulin acetyltransferase (MEC-17) and αK40-specific deacetylation (*K40R*) (*Akella et al., 2010*) by a combination of cryo-EM, molecular dynamics and mass spectrometry.

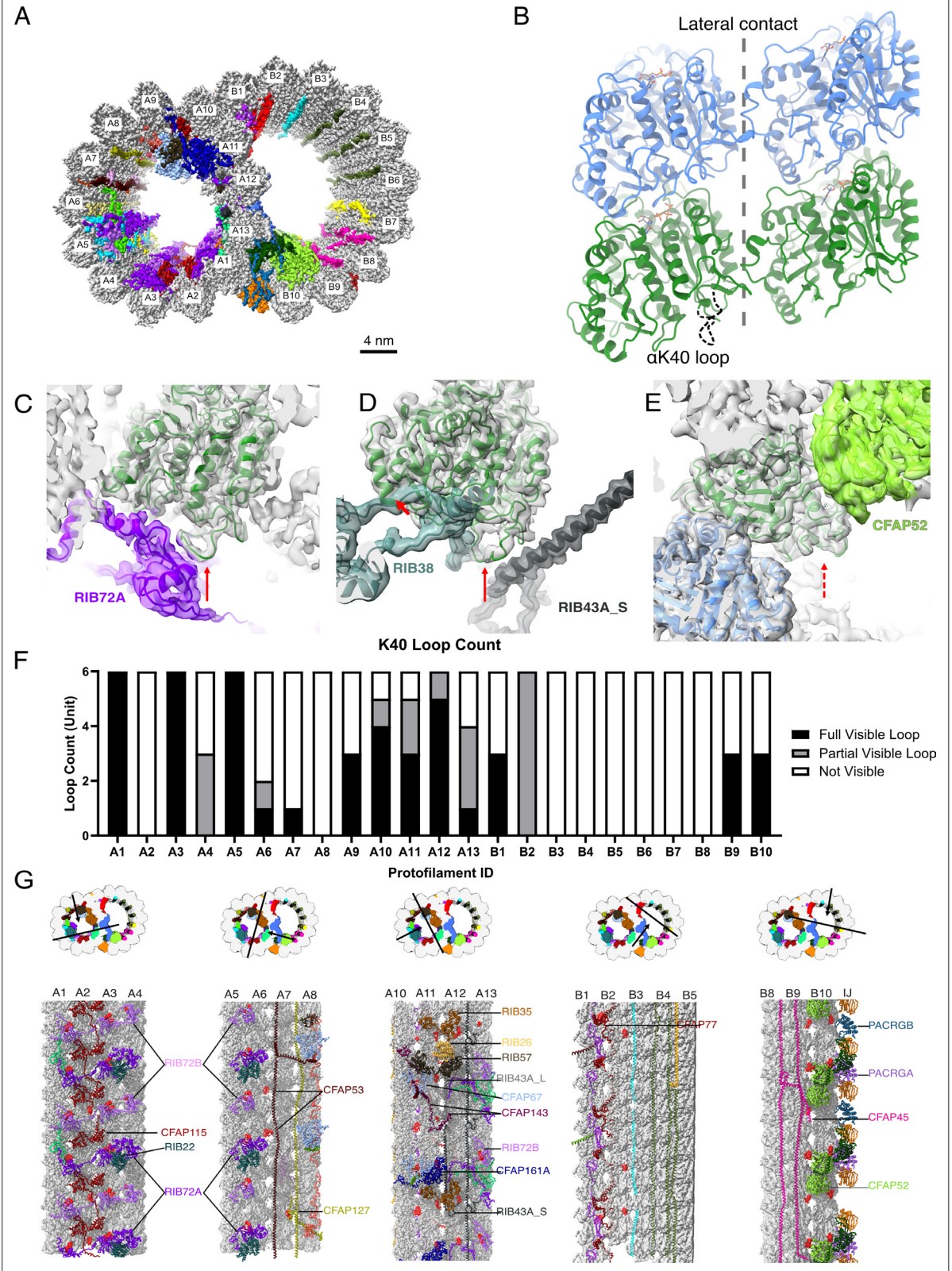

**Figure 1.** The presence of the structure αK40 loops in *Tetrahymena thermophila* doublet microtubule (DMT). (**A**) Surface rendering of the DMT viewed from the tip of the cilia, with microtubule inner proteins (MIPs) colored in the 48nm repeat cryo-EM electron density map of *Tetrahymena*. (**B**) Relative location of the αK40 loop (dashed black line) and the lateral contacts of tubulins. Color: α-tubulin, green; β-tubulin, blue. Cryo-EM map and models of the fully structured αK40 loops in protofilament (PF) A3 (**C, D**) and the fully structured (**E**) and partially structured αK40 loops in PF B10. The full

*Figure 1 continued on next page*

*Figure 1 continued*
red arrows point to the location of the αK40 loops, and dashed red arrow points to the location of the partial αK40 loops. (**F**) Bar graph showing the composition of visible full (missing no more than two residues from residues 37 to 48) and partial loops (missing 3–5 residues) in both the A- and B-tubules. (**G**) 48nm repeat surface renderings of selected PFs with MIPs that interact with visible full and partial αK40 loops colored in red, indicating that αK40 loops are structured in regions with many MIPs.

The online version of this article includes the following figure supplement(s) for figure 1:

**Figure supplement 1.** Purification and structure determination of doublet microtubule (DMT).

**Figure supplement 2.** Diverse structures of the αK40 loops.

## Results

## Acetylated αK40 loops are structured when interacting with certain MIPs

We first examined a 48 nm repeat cryo-EM density map reconstructed from native DMTs of *Tetrahymena WT* (*CU428* strain) (*Figure 1A*, *Figure 1—figure supplement 1*, *Tables 1 and 2*; *Kubo et al., 2023*). We identified and modeled all visible αK40 loops of α-tubulins in the map, which are known to be almost 100% acetylated (*Akella et al., 2010*; *Gaertig et al., 1995*). Most *Tetrahymena* MIPs have been localized and identified in this map (*Kubo et al., 2023*). In contrast to the acetylated singlet microtubule structure (*Eshun-Wilson et al., 2019*), we observed many fully structured αK40 full (missing no more than two residues from residues 37 to 48) and partial structure loops (missing 3–5 residues) in DMT (*Figure 1C–E*). In certain PFs, including A1, A3, A5, and A12, all αK40 loops are structured. Interestingly, the structured αK40 loops are much less abundant in the B-tubule (*Figure 1F*). There is a pattern in the distribution of αK40 loops: likely due to their flexibility, ordered αK40 loops are visible at positions where MIPs and tubulins interact but are difficult to resolve in places with little or no MIPs (*Figure 1G*, *Figure 1—figure supplement 2*, *Supplementary file 1a*). Notably, in A1, where RIB72A and RIB72B are in contact with αK40 (*Figure 1C*, *Figure 1—figure supplement 2*), all αK40 loops are fully structured, in agreement with the 8 nm alternating pattern of Rib72A and Rib72B (*Figure 1G*). This pattern is consistent with previous observations of the αK40 loops in the green algae *C. reinhardtii* DMT (*Khalifa et al., 2020*; *Ma et al., 2019*). Therefore, the B-tubule has fewer ordered αK40 loops compared to the A-tubule, probably due to fewer MIPs in the B-tubule.

A superimposition of the conformations of all fully structured αK40 loops in the DMT revealed that they adopt multiple conformations (*Figure 2A*). The exact structures of αK40 loops might adapt to

**Table 1.** Cryo-EM data collection and processing parameters for all datasets used in this study.

| | WT (CU428) | | K40R | | MEC17-KO |
|---|---|---|---|---|---|
| Dataset | Dataset 1 | Dataset 2 | Dataset 1 | Dataset 2 | Dataset 1 |
| Microscope | Titan Krios | Titan Krios | Titan Krios | Titan Krios | Titan Krios |
| Electron detector | Gatan K3 | Gatan K3 | Gatan K3 | Gatan K3 | Gatan K3 |
| Zero-loss filter (eV) | 30 | 30 | 30 | 30 | 30 |
| Magnification | 64,000 | 64,000 | 64,000 | 64,000 | 64,000 |
| Voltage | 300 | 300 | 300 | 300 | 300 |
| Electron exposure | 45 e/A$^2$ | 45 e/A$^2$ | 73 e/A$^2$ | 45 e/A$^2$ | 45 e/A$^2$ |
| Defocus range | 1.0–3.0 μm | 1.0–3.0 μm | 1.0–3.0 μm | 1.0–3.0 μm | 1.0–3.0 μm |
| Pixel size | 1.37 | 1.37 | 1.37 | 1.37 | 1.37 |
| Symmetry imposed | C1 | C1 | C1 | C1 | C1 |
| Movies acquired | 4080 | 14,304 | 4400 | 21,210 | 4283 |
| Particles number | 40,945 | 125,306 | 30,352 | 152,303 | 39,417 |
| Global resolution (Å) | 4.51 | 4.26 | 4.0 | 3.6 | 4.5 |
| Local resolution (Å) | 3.6–4.0 | | 3.3–3.5 | | 4.0–4.5 |

**Table 2.** Refinement statistics of *WT*, *K40R*, and *MEC17-KO* 48 nm models.

| | WT (CU428) | K40R | MEC17-KO |
|---|---|---|---|
| Model-to-map fit, CCmask | 0.8009 | 0.7961 | 0.6070 |
| All-atom clashscore | 16.61 | 13.21 | 57.06 |
| Ramachandran plot | | | |
| Outliers [%] | 0.14 | 0.14 | 0.14 |
| Allowed [%] | 3.70 | 3.60 | 6.13 |
| Favored [%] | 96.16 | 96.26 | 93.73 |
| Rotamer outliers [%] | 0.03 | 0.04 | 0.02 |
| Cbeta deviations [%] | 0.00 | 0.00 | 0.01 |
| Cis-proline [%] | 4.56 | 4.56 | 4.54 |
| Cis-general [%] | 0.01 | 0.01 | 0.03 |
| Twisted proline [%] | 0.17 | 0.18 | 0.13 |
| Twisted general [%] | 0.02 | 0.01 | 0.02 |

Supplementary data for cryo-EM observation and mass spectrometry.

both the inter-PF angle and interactions with the contacting MIPs. We did not observe any conformations where acetylated αK40 was directly involved in tubulin–tubulin lateral interactions (*Figure 1C–E* and *Figure 2A*).

We noticed that the αK40 loop is always structured when interacting with DM10 domains of the MIPs, suggesting that DM10 is a αK40 loop interacting domain. Superimposing all seven DM10 domains (three from RIB72A, three from RIB72B, and one from CFAP67) (*Figure 2B*) showed that while there are some variations in other parts of DM10 domain, the region interacting with the αK40 loop are conserved in its topology (*Figure 2B, C*). Sequence alignment of the DM10 domains suggests that an aromatic residue (phenylalanine or tyrosine) is conserved and might be responsible to the interaction with the DM10 domain (*Figure 2C–E*).

To determine whether the conformation of the αK40 loops is conserved between species, we compared the αK40 loops in the DMTs of *T. thermophila* and *C. reinhardtii* (*Figure 2F–H*). The αK40 loop conformations from B9 and B10 are similar between the two species, and both interact with the conserved CFAP52. These results suggest that the αK40 loop conformation is tuned by its interactions with adjacent MIPs. The interaction between MIPs and the αK40 loop implies that acetylation might play a role in DMT assembly or stability or both.

## αK40 acetylation stabilizes the inter-PF angles in the B-tubule

With our observation that acetylated αK40 in the DMT of *WT Tetrahymena* cells adopts a fixed conformation when interacting with MIPs, we evaluated whether the absence of αK40 acetylation affects the MIPs and hence the overall DMT structure. We obtained the DMT structures from the *Tetrahymena MEC17-KO* and *K40R* mutants at 4.5 and 3.5 Å resolution, respectively (*Tables 1 and 2*). In *MEC17-KO*, *MEC17* – an ortholog of the mammalian *αTAT1* α-tubulin K40 acetyltransferase, is knocked out to abolish detectable αK40 acetylation, while in the *K40R* mutant, lysine 40 on ATU1 (the single canonical α-tubulin isotype in *Tetrahymena*) is mutated to arginine to prevent acetylation at the αK40 position (*Akella et al., 2010*).

Our first observation is that all the decoration of the MIPs appears intact in *MEC17-KO* and *K40R* DMT compared to that of *WT* (*Figure 3A*, *Figure 3—figure supplement 1A, B*). This observation is consistent with the phenotypes that the *MEC17-KO* and *K40R* cilia look similar to the *WT* cilia (*Akella et al., 2010*; *Gaertig et al., 1995*). Therefore, our cryo-EM analyses indicate that acetylation does not affect DMT and MIP assembly.

Next, we examined the tubulin structure by comparing the αK60 and αK40 loop structures at PFs A3 and B1 (*Figure 3B–I*). Previously, it was reported that αK40 acetylation led to a reduction in the distance of αK60 from the M-loop of the adjacent tubulin from 12 to 8 Å (*Eshun-Wilson et al., 2019*). Our comparison of (acetylated and non-acetylated) αK40 and αK60 loops did not reveal significant structural differences. However, this result does not rule out a change in αK40 and αK60 conformation, which is likely too small to be observed accurately at this resolution.

Since αK40 loops are suggested to be involved in lateral interactions (*Cueva et al., 2012*; *Eshun-Wilson et al., 2019*) and acetylated αK40 results in weakened inter-PF interactions (*Portran et al., 2017*), acetylation might affect inter-PF angles. We measured the inter-PF angles in the DMTs of the

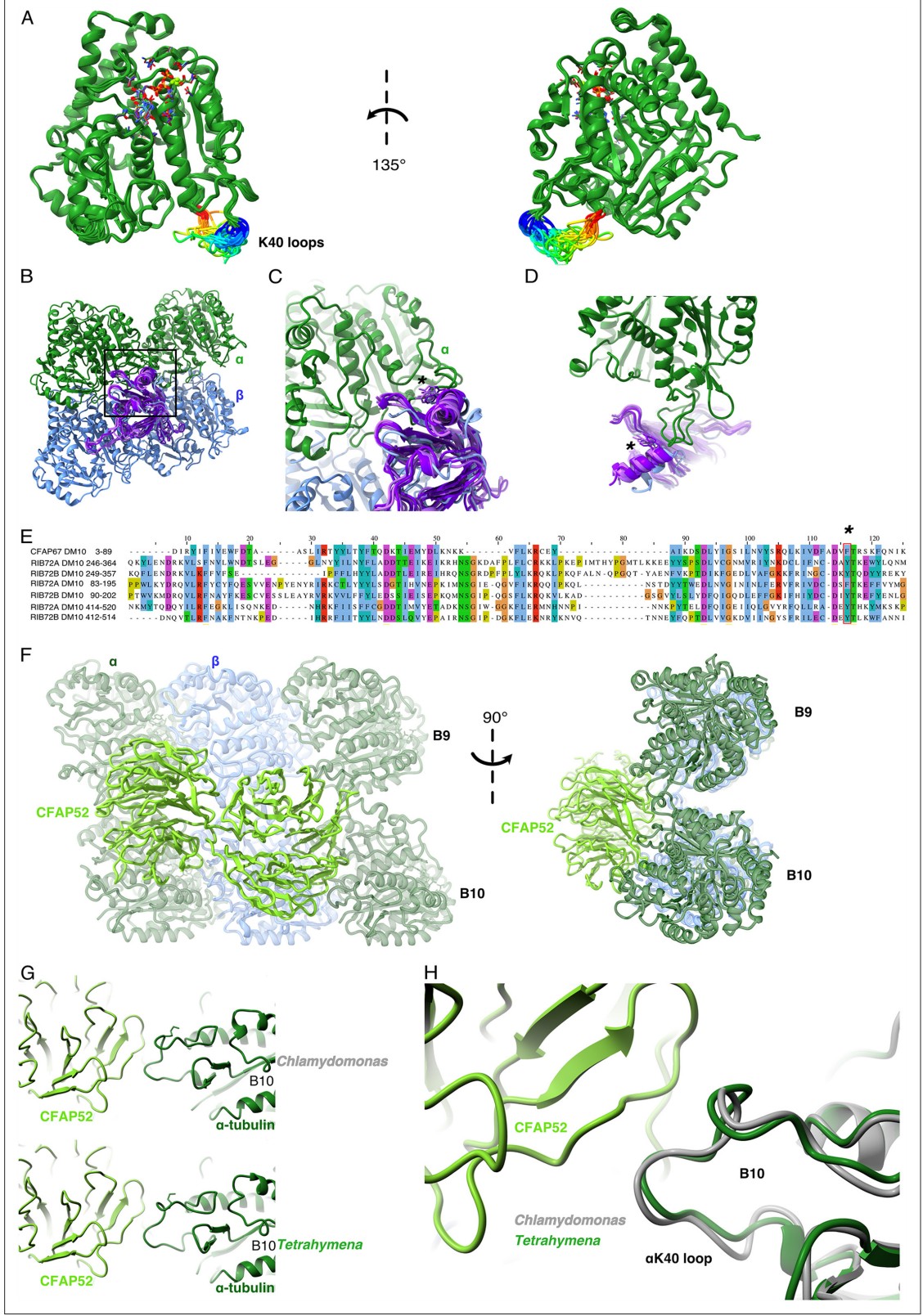

**Figure 2.** Comparison of αK40 loop conformation. (**A**) Superimposed view of all the orientations of all the visible full and partial αK40 loops, showing their orientation. (**B**) Interaction of αK40 loop and DM10 domains from RIB72A (three domains, blue), RIB72B (three domains, purple), and CFAP67 (one domain, cyan). Black box represents the view in (**C**). (**C**) Zoom in view of DM10 domains and αK40 loop interaction. Asterisk (*) denotes the conserved aromatic residue potentially interact with the αK40 loop. (**D**) Alternative view of DM10 domains interacting with αK40 loop. (**E**) Multiple sequence

*Figure 2 continued on next page*

*Figure 2 continued*

alignment of DM10 domains from RIB72A, RIB72B, and CFAP67. (**F**) Cryo-EM map (left) and model (right) of the inner junction region of *Tetrahymena* to show the interaction of the full αK40 loop with CFAP52. (**G**) Cryo-EM map (left) and model (right) of the inner junction region of *Chlamydomonas* to show the interaction of the full αK40 loop with CFAP52. (**H**) Superimposed view of the *Chlamydomonas* αK40 loop (gray) onto the *Tetrahymena* αK40 loop of B10.

*WT* and the two acetylation-deficient mutants. We observed only minor changes in the inter-PF angles in the A-tubule (*Figure 4A*, *Figure 4—figure supplement 1A*). However, in the B-tubule, the changes in the inter-PF angles were more prominent (*Figure 4A–M*, *Figure 4—figure supplement 1B*). Most notably, the change in the inter-PF angle between PFs B7 and B8 and B9 and B10 ranges from 3° to 6° (*Figure 4H–M*, *Figure 4—figure supplement 2*, *Supplementary file 1b*). These are significant changes in specific PF curvatures. To make sure that our observation is statistically significant, we analyzed the inter-PF angles using independent reconstructions obtained from two biological replicates of the *WT* and *K40R* DMT. The measurements of inter-PF angles between the two *WT* replicates are consistent, suggesting that the inter-PF angles in *WT* are well controlled. For two *K40R* replicates, the inter-PF angles have minor changes in the A-tubule (*Figure 4—figure supplement 1A*) but for the B-tubule, there are several PF pairs with significant differences (*Figure 4—figure supplement 1B*), especially in PFs B7B8 and B9B10. This analysis suggests that the inter-PF rotation angle varies significantly more in acetylation mutants *K40R* and *MEC17-KO* (*Figure 4—figure supplement 1B*).

We also inspected the tubulin lattice by measuring the distances between adjacent tubulin dimers. There were small changes in the interdimer distances in PFs A1, A3, A5, A8, B3, and B7 between *WT* and mutants (*Figure 4—figure supplement 1C*, *Supplementary file 1c*). This scale of changes in the tubulin lattice is significantly smaller (0.3–0.5 Å) than the known impacts of the nucleotide state (*Zhang et al., 2015*) and missing MIPs (*Ichikawa et al., 2019*) (~2 Å difference). This is similar to the observation of differences in interdimer distances in 96% acetylated and 99% deacetylated microtubules using in vitro reconstitution.

Overall, our analysis shows that tubulin lattice changes, specifically inter-PF angles, can be used as an indication of microtubule stability when acetylation is lacking.

## Acetylated αK40 loops are less flexible

While we observed some structural differences between acetylated and non-acetylated DMT, our resolution does not allow us to see small changes in the αK40 loops. Therefore, we attempted to detect structural differences using molecular dynamic simulations.

We performed an all-atom molecular dynamic simulation for α-tubulin with acetylated and non-acetylated αK40 to determine whether acetylation changes the loop behavior (*Figure 5A, B*). To determine how acetylation of αK40 affects the structure of the αK40 loop, we used the *K*-means clustering method for the αK40 loop region based on the Cα position. To avoid bias in clustering, the structures obtained during the simulated trajectories with acetylated and non-acetylated αK40 were mixed, and 10 clusters were created (*Figure 5A, B*). Interestingly, clusters comprise mostly acetylated or non-acetylated conformations but not mixed populations. This indicates that acetylated and non-acetylated αK40 loops adopt distinctively different conformations. In addition, the most populated cluster consists of ~50% of all acetylation conformations, while the non-acetylated αK40 loops form five clusters of less than 30% conformations. Therefore, the acetylated αK40 loop adopts more rigid conformations than non-acetylated αK40 loop. It appears that this finding is similar to that of a previous study (*Eshun-Wilson et al., 2019*) despite differences in molecular dynamic setups of porcine and ciliate tubulins and clustering methods.

## Lack acetylation of αK40 does not significantly affect tubulin and MIP interactions

Since we did not observe any obvious structural changes in the decoration of the MIPs in the cryo-EM maps of the 48 nm repeat of *MEC17-KO* and *K40R* mutants compared to *WT* (*Figure 3A*, *Figure 3—figure supplement 1*), non-acetylated αK40 does not appear to significantly weaken the binding of MIPs. Therefore, we further explored the degree to which the interaction between the αK40 loop and MIP is affected by acetylation using coarse grain molecular dynamic simulation to compare the

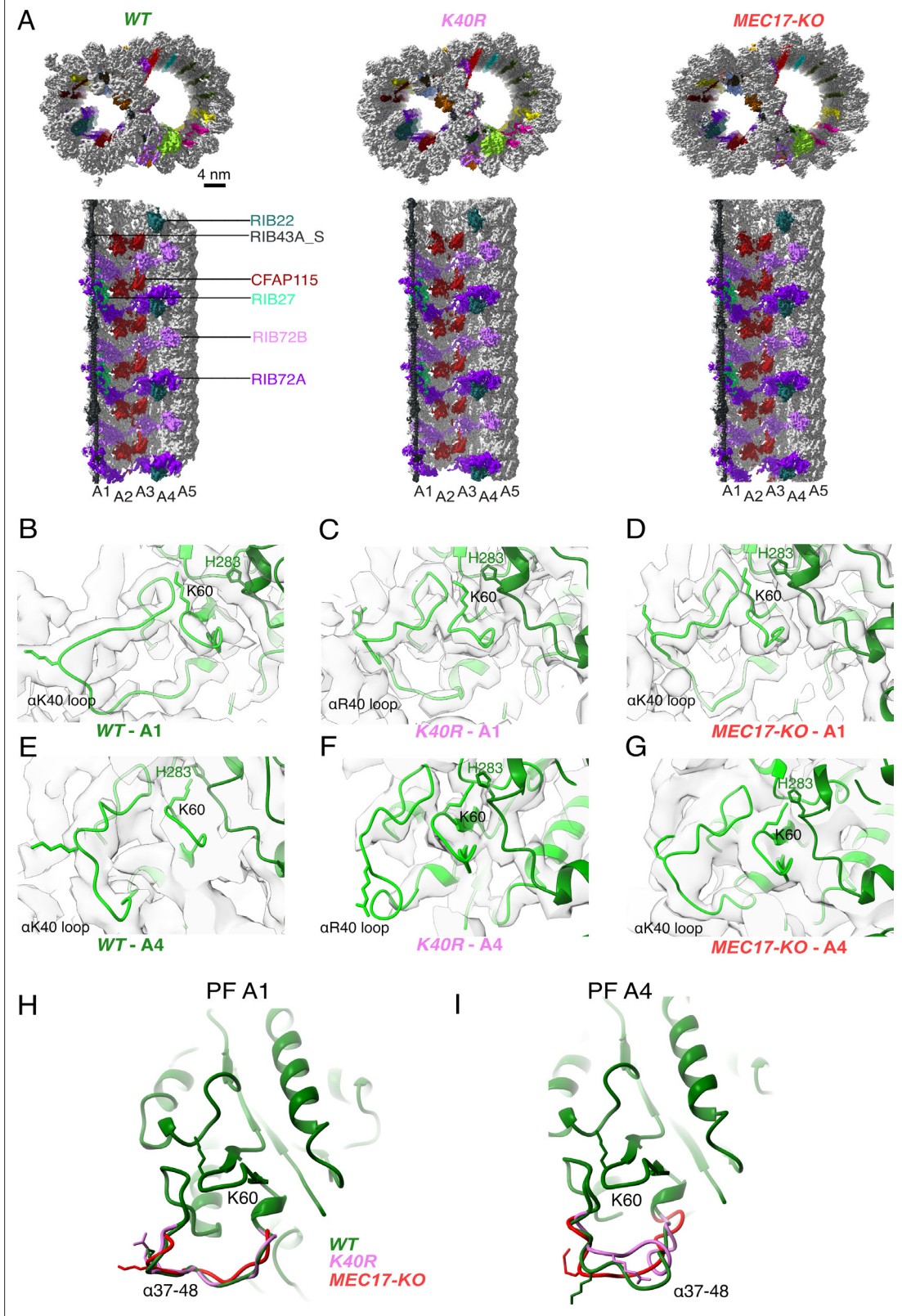

**Figure 3.** Comparison of doublet microtubule (DMT) structures from *WT*, *MEC17-KO*, and *K40R* mutants. (**A**) Comparison of the cryo-EM density maps of the DMT from *WT*, *K40R*, and *MEC17-KO* strains of *Tetrahymena* to show that the microtubule inner proteins (MIPs) are intact in all three species. Models of the full αK40 loops in protofilament (PF) A1 (**B–D**) and PF A4 (**E–G**) from *WT*, *K40R*, and *MEC17-KO* strains. Models of the αK40 loops from A1 (**H**) and A4 (**I**) superimposed from *WT*, *K40R*, and *MEC17-KO* species.

*Figure 3 continued on next page*

*Figure 3 continued*

The online version of this article includes the following figure supplement(s) for figure 3:

**Figure supplement 1.** Decoration of microtubule inner proteins (MIPs) are similar in 48 nm cryo-EM maps of the doublet microtubule (DMT) from *WT*, *K40R*, and *MEC17-KO*.

energy between tubulins and CFAP52 with the acetylated αK40 and the non-acetylated αK40 structures (*Figure 5C*).

We found that both acetylated and non-acetylated αK40 have a stabilizing effect when binding to CFAP52. On the other hand, when calculating the mean and standard error of the energy between tubulins and CFAP52 from the entire trajectory, the energy of the acetylated αK40 case is only slightly lower than that of the non-acetylated αK40 case (*Figure 5D*). Similarly, no significant difference between acetylated and non-acetylated αK40 in interactions with similar setup using RIB72A. Even when accounted for the periodic structures of DMT with many MIPs, the energy difference between acetylated and non-acetylated αK40 with MIPs is still not significant.

Our coarse grain molecular dynamic simulations of MIPs and αK40 suggest that acetylation likely does not play a significant role in the interaction with MIPs.

## Mass spectrometry reveals changes in DMT protein composition in response to a lack of αK40 acetylation

Our analysis thus far has failed to reveal significant structural differences in the MIPs within the DMT structures of *WT*, *K40R*, and *MEC17-KO* mutants. Therefore, we searched for more subtle changes in the axoneme composition using mass spectrometry. We analyzed the same DMT samples used for cryo-EM, which contain no membrane and matrix fractions, to see any proteomic changes due to the lack of acetylation. In addition, to eliminate the downstream effect of the lack of acetylation that is destabilization of DMT, we combined the mass spectrometry results in this study with the mass spectrometry studies of the *RIB72A/B-KO* and *RIB72B-KO* mutants (*Kubo et al., 2023*). The *RIB72A/B-KO* mutant lacks both RIB72A and RIB72B, which leads to a significant number of MIPs missing and slower swimming speed (*Stoddard et al., 2018*). Therefore, we can use *RIB72B-KO* and *RIB72A/B-KO* as controls to specifically look for the upstream effect due to the lack of acetylation.

We first performed a control of proteins not supposed to interact with αK40 by analyzing the levels of radial spoke proteins and found no significant differences (*Figure 6—figure supplement 1A*). We then looked at the MIP abundance. Most MIPs showed no significant changes in their abundance (less than 1.5-fold changes) except for CFAP112, CFAP141, and RIB27 (*Figure 6A*) in the *K40R* and *MEC17-KO* samples. However, we did not observe any differences in the periodicity of CFAP112, CFAP141, and RIB27 in the cryo-EM maps of the DMTs from *WT*, *K40R*, and *MEC17-KO* mutants. As shown recently in the case of CFAP77A and CFAP77B, certain MIPs might not localize consistently along the length of the cilium (*Kubo et al., 2023*), and there might be some changes in the occupancy of those MIPs in a specific region that lead to changes in their abundance.

There were 11 proteins significantly elevated in both *K40R* and *MEC17-KO* mutants by fourfold compared to the *WT* cilia (*Figure 6B*, *Figure 6—figure supplement 1B, C*, *Supplementary file 1d, e*). One of these proteins was also elevated in the *RIB72A/B-KO* mutant vs. *WT*, and therefore, 10 proteins were specifically upregulated in both the *K40R* and *MEC17-KO* strains (*Figure 6B, G*). Similarly, we found that three proteins were significantly reduced in both *K40R* and *MEC17-KO* but not in the *RIB72A/B-KO* knockout mutant (*Figure 6C, E* and *Supplementary file 1f, g*). Among the proteins reduced or missing in the *K40R* and *MEC17-KO* mutants, there are protein phosphatase 2A-related proteins: PP2A regulatory subunit A (TTHERM_00766530, UniProt ID: I7MAR7), PP2C (TTHERM_00316330, UniProt ID: I7LW71), and PP2A (TTHERM_00355160, UniProt ID: Q22Y55) (*Figure 6D*). Furthermore, one kinase was downregulated in *MEC17-KO* and *K40R* cells (*Figure 6E*) (TTHERM_00623090, UniProt ID: Q240X5). In *Chlamydomonas*, PP2A is present in DMT and required for normal ciliary motility (*Elam et al., 2011*). It was previously reported that the knockdown of PPP1R2 (protein phosphatase inhibitor 2) reduces αK40-acetylation in the primary cilium of human retinal epithelial cells (*Wang and Brautigan, 2008*). Inhibition of protein phosphatase (1 and 2 A) with calyculin in PPP1R2 knockdown cells partially rescued the acetylation of ciliary microtubules. These

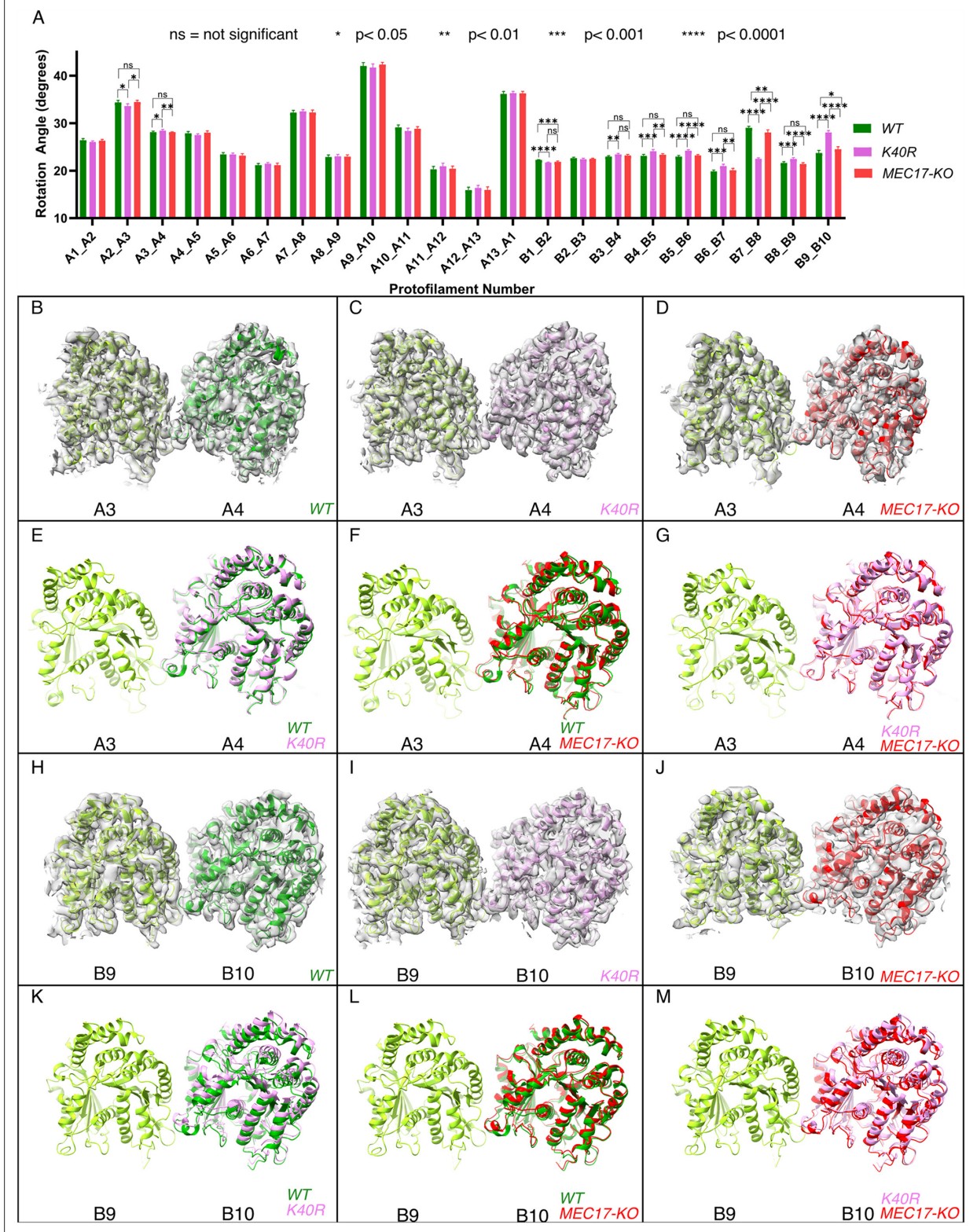

**Figure 4.** Deacetylation affects the inter-protofilament (PF) angles in the doublet microtubule (DMT). (**A**) Inter-PF rotation angles for each PF across all three strains (*WT*, *K40R*, and *MEC17-KO*) measured from six different pairs of tubulin dimers along the 48 nm repeat of the DMT. (**B–G**) Comparison of inter-PF rotation angle change between A3 and A4, showing minor changes. (**H–M**) Comparison of inter-PF rotation angle change between B9 and B10, showing significant changes.

The online version of this article includes the following figure supplement(s) for figure 4:

*Figure 4 continued on next page*

Figure 4 continued

**Figure supplement 1.** Tubulin lattice measurement from the doublet microtubule (DMT) from *WT*, *K40R*, and *MEC17-KO Tetrahymena* species.

**Figure supplement 2.** Deacetylation affects the inter-protofilament (PF) angles in the doublet microtubule (DMT).

results suggest that the lack of αK40 acetylation reduces PP2A activity in DMT. Therefore, we can infer that acetylation and phosphorylation interact in DMT.

## Discussion

In this work, we reinforced the notion that acetylation of αK40 makes the loop less flexible (*Eshun-Wilson et al., 2019*), which is observed by molecular dynamics. In addition, we performed structural characterization of the DMTs from two acetylation mutants, *K40R* and *MEC17-KO*, and compared them to that of the *WT*. We showed that the αK40 loops are structured when they interact with MIPs and identified DM10 as an αK40 interacting domain. This suggests that the αK40 loop plays an

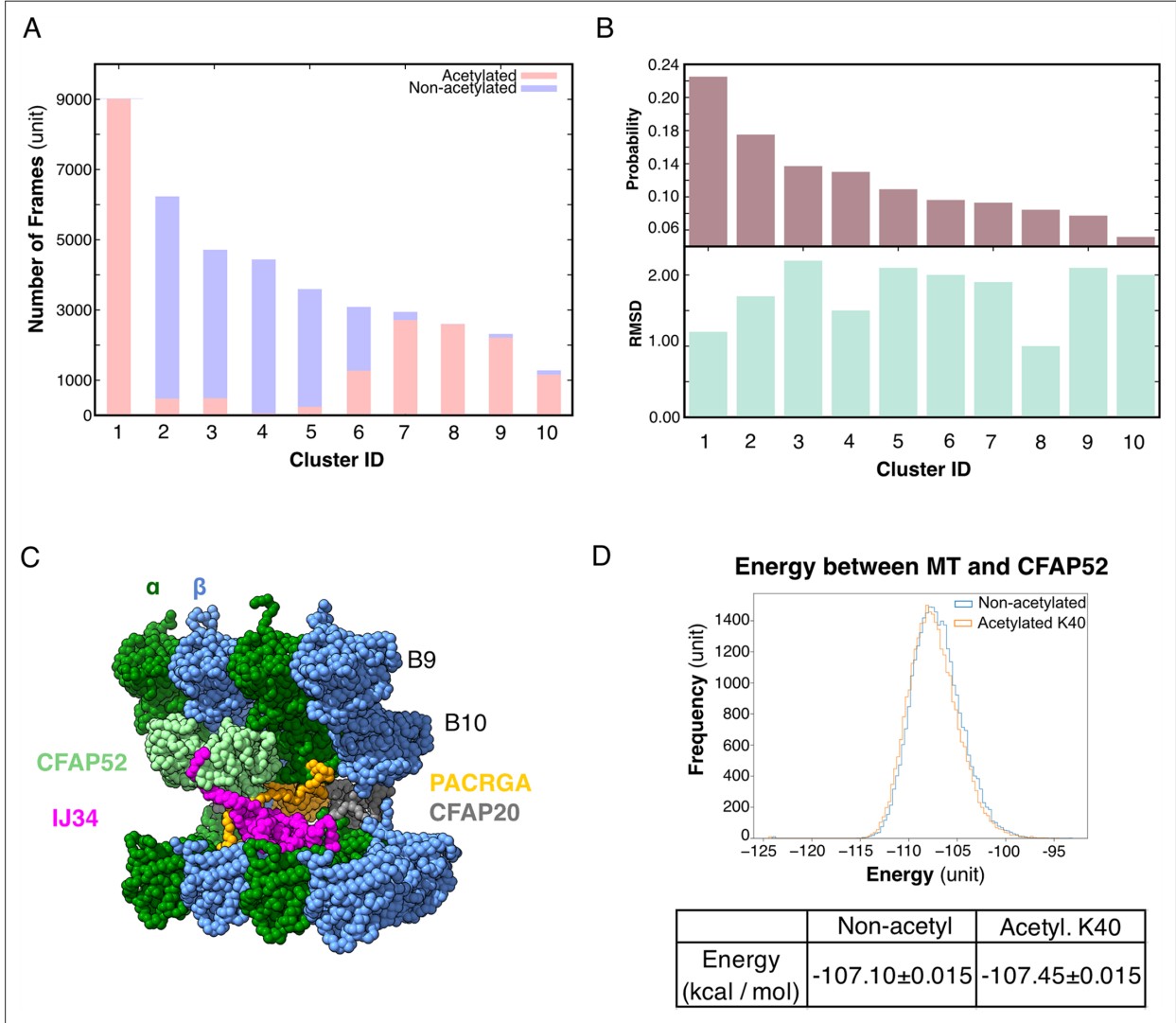

**Figure 5.** Molecular dynamic simulations of the acetylated and non-acetylated αK40 loops. (**A**) All-atom simulations of αK40 loop clusters in different conformations of acetylated (pink) and base/non-acetylated (blue) indicate that acetylated conformations adopt higher frames and are less flexible. (**B**) Root Mean Square Deviation (RMSD) and probability of each cluster simulated in A. (**C**) Molecular dynamics coarse grain model of the inner junction region of *Tetrahymena*; each amino acid is 1 bead. (**D**) Graph showing the energy difference (in kcal/mol) between base (non-acetylated) and acetylated αK40 to show that each acetylated αK40 has slightly lower energy than the non-acetylated αK40.

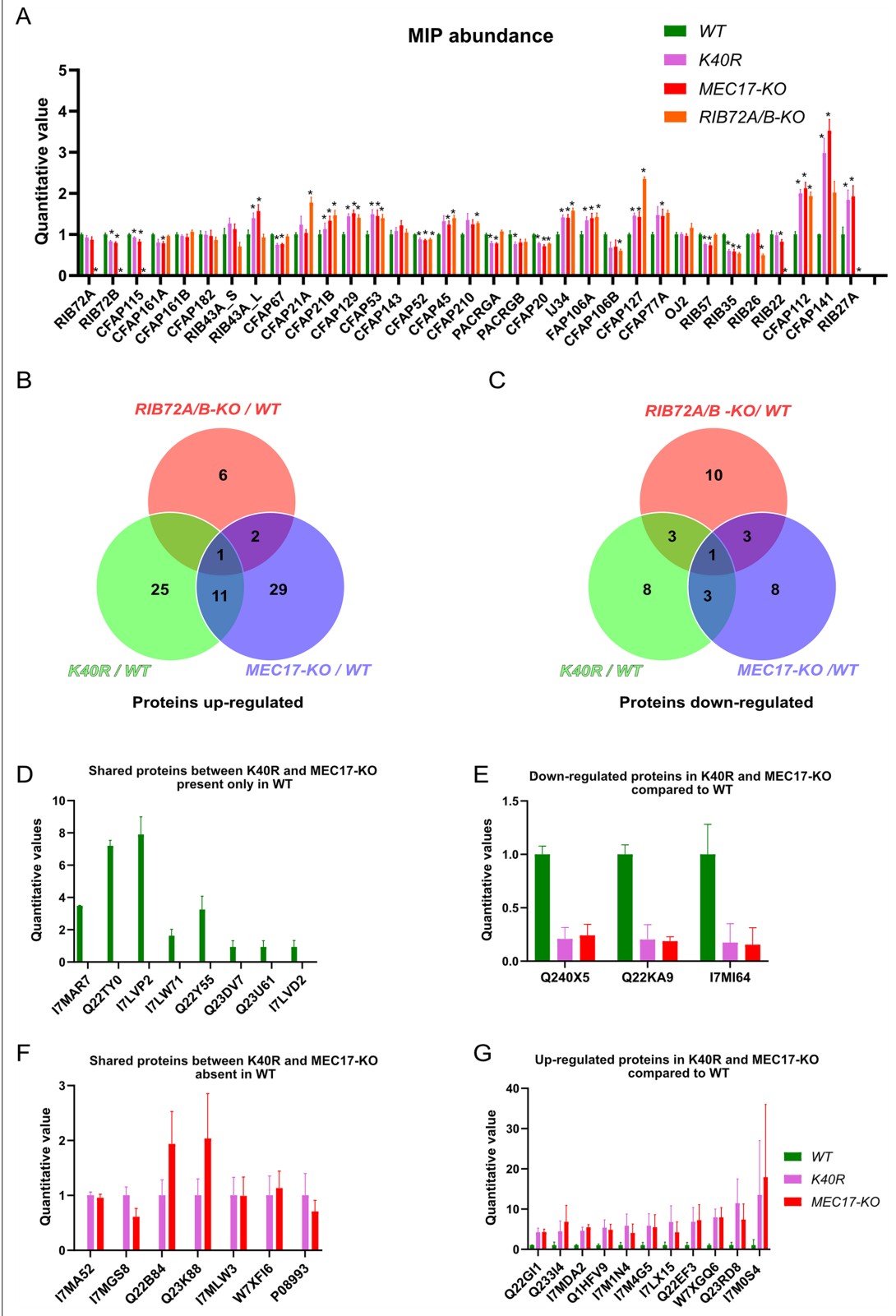

**Figure 6.** Mass spectrometry of *WT*, *K40R*, and *MEC17-KO* mutants. (**A**) Bar graph showing the abundance of microtubule inner proteins (MIPs) based upon quantitative values (normalized total spectra) from mass spectrometry (n = 3 biological replicates). Asterisk (*) indicates a significant difference with p < 0.05. (**B**) Proteins upregulated in *RIB72A/B*, *K40R*, and *MEC17-KO* mutants compared with the *WT*. (**C**) Proteins downregulated in *RIB72A/B*, *K40R*, and *MEC17-KO* mutants compared to the *WT*. (**D**) Proteins only found in the mass spectrometry of *WT* when compared with *K40R* and *MEC17-KO*

*Figure 6 continued on next page*

*Figure 6 continued*

mutants. (**E**) Downregulated proteins in both *K40R* and *MEC17-KO* mutants compared to *WT*. (**F**) Proteins in both *K40R* and *MEC17-KO* mutants but are absent in *WT*. (**G**) Upregulated proteins in both *K40R* and *MEC17-KO* mutants compared to *WT*.

The online version of this article includes the following figure supplement(s) for figure 6:

**Figure supplement 1.** Mass spectrometry analysis of *WT*, *K40R*, and *MEC17-KO* (**A**) Radial spoke protein abundance in mass spectrometry triplicates of *WT*, *K40R*, and *MEC17-KO Tetrahymena* species (n = 3 biological replicates).

important role in certain MIP–tubulin interactions. On the other hand, the αK40 loop–MIP interactions do not detectably change their structures between acetylated and non-acetylated αK40. These results imply that a complete DMT with MIPs is assembled without the need of acetylation. Later, the MEC-17/αTAT1 acetyltransferase acetylates αK40 within the DMT.

Interestingly, we observed significant changes in inter-PF angles in the B-tubule, where MIPs are fewer. Our results suggest that αK40 loop interactions with MIPs are the dominant interactions that stabilize microtubules regardless of acetylation status. When there are fewer MIPs and thus less interaction between the αK40 loop and MIPs, the contribution of the acetylation of αK40 to the lateral interaction between adjacent PFs becomes more significant (*Figure 7*). As a result, the lack of acetylation destabilizes DMT and leads to tubulin lattice alteration and instability. Our results agree with an in vitro study showing that αK40 acetylation affects inter-PF interaction and, as a result, improves mechanical properties of reconstituted microtubules (*Portran et al., 2017*). In a recent study (*Viar et al., 2023*), the tip region of the *Chlamydomonas* cilia is acetylated early during growth compared to other PTM such as polyglutamylation and polyglycylation. In the tip, the microtubules exist as singlet microtubules and do not have regular MIP binding pattern like in the base (*Legal et al., 2023*). As a result, acetylation might play an essential role in the ciliary tip region to stabilize microtubules. Moreover, our study supports the notion that acetylation of tubulin is not a biphasic switch but a fine-tuning mechanism that impacts microtubule stability. Our results again demonstrate that the tubulin lattice can be a read-out for DMT stability (*Ichikawa et al., 2019*). Recently, it has been shown that acetylation of K394, which is located at the αβ-tubulin dimer interface, is specific to flies' nervous system, and is critical to neuronal growth (*Saunders, et al., 2022*). This finding might suggest that different acetylation sites can be fine-tuned for different purposes.

Since both acetylation mutants lack protein phosphatase 2A and its regulatory subunits, there could be an interaction between tubulin acetylation and protein phosphatase 2A in cilia. TGF-β-activated kinase 1 (TAK1) is an important activator of αTAT1 in mice (*Shah et al., 2018*). We can speculate that the absence of αK40 acetylation could also affect the enzyme MEC-17/αTAT1 and its availability for phosphorylation by TAK1 or other kinases. Possibly, the lack of αK40 acetylation changes the

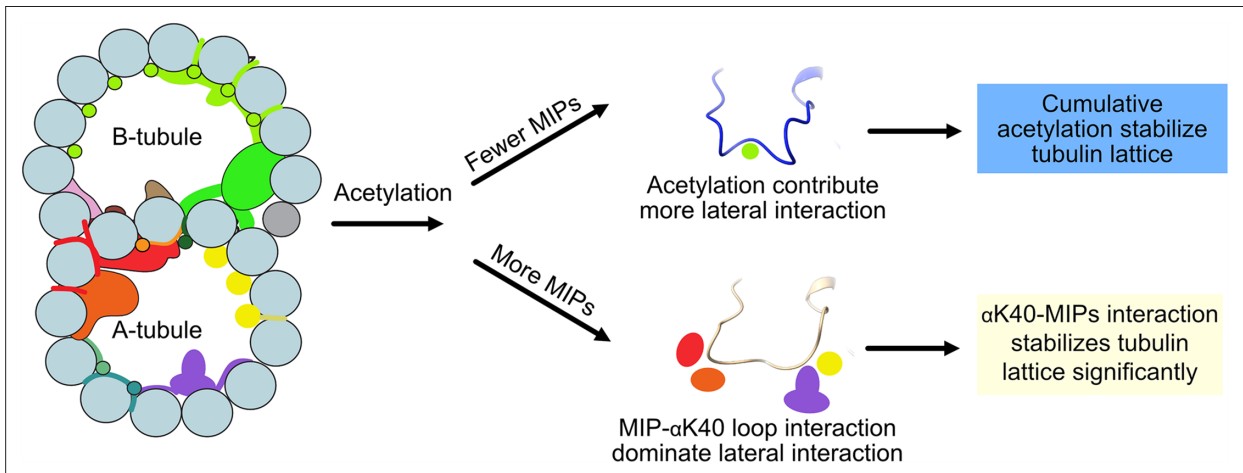

**Figure 7.** Models of acetylation contribution in the DMT In the case of more microtubule inner proteins (MIPs), such as in the A-tubule, the MIP–αK40 interaction dominates the contribution to the lateral interaction; therefore, deacetylation does not affect the structures significantly. With fewer MIPs, such as in the B-tubule, acetylation contribution to the lateral interaction becomes significant and therefore can contribute to the stabilization of the tubulin lattice.

phosphorylation levels by altering the balance between kinases and phosphatases (protein phosphatase 2A in this case). Interestingly, HDAC6 deacetylase is also regulated by multiple kinases (*Du et al., 2015*). This again highlights the interdependence of phosphorylation and acetylation in cilia. Furthermore, crosstalk occurs between αK40 acetylation and other tubulin PTMs, namely polymodification. For example, *Tetrahymena* mutants lacking tubulin polyglycylases (TTLL3) have elevated levels of αK40 acetylation (*Wloga et al., 2009*). Our mass spectrometry results showed that a tetratricopeptide repeat protein (TTHERM_00313720, UniProt ID: Q22KA9) was downregulated in *MEC17-KO* and *K40R* mutants (*Figure 6E*). Protein BLAST found a homolog in *Homo sapiens* as tetratricopeptide repeat protein 30B/IFT70 (UniProt ID: Q8N4P2), an IFT protein whose depletion reduces polyglutamylation of axonemal tubulin (*Pathak et al., 2007*). We also observed that the absence of αK40 acetylation affects the PF angles, and the impact is stronger on the B-tubule. The B-tubule is enriched in polymodification (polyglutamylation and polyglycylation), and therefore, the altered B-tubule structure could affect the access of enzymes that add or remove polymodification. In fact, some of the effects of the absence of αK40 acetylation based on the use of non-acetylated axonemes (*Reed et al., 2006*) could be due to alterations of the B-tubule, namely, the levels of polymodification. Together with our results and others (*Akella et al., 2010*; *Kubo et al., 2015*; *Rogowski et al., 2010*; *Wloga et al., 2008*; *Wloga et al., 2009*), these results suggest that the PTM in cilia is in a balancing act. Changing one type of PTM can shift the balance of other types of PTMs.

## Materials and methods

### Growth of *Tetrahymena* cells for isolation

Bean media (*Williams et al., 1980*) was used to store *Tetrahymena* cells (*WT* (*CU428*), *K40R*, and *MEC17-KO*), with 4 μl of the cell culture transferred into 40 ml of liquid Super Proteose Peptone (SPP) media (*Gorovsky et al., 1975*). The cells were grown for 7 days at room temperature and then transferred into 50 ml of SPP liquid media overnight growth at 30°C with 150 RPM shaking in a Thermo Fisher MAXQ8000 incubator. Then, 50 ml of this overnight culture was added to 900 ml of liquid SPP media and grown at 30°C (MAXQ8000) at 150 RPM for 2 days or until the $OD_{600}$ was 0.7.

### Flagella isolation via dibucaine treatment

To harvest *Tetrahymena* cells, overnight culture was centrifuged at 700 × *g* for 10 min with slow deceleration at 22°C in an Avanti Centrifuge (Rotor JLA-8.1000). Ten milliliters of room temperature SPP medium supplemented with dithiothreitol (DTT) was used to resuspend the cell pellet and then adjusted to a final volume of 24 ml, followed by transfer to an ice-cold 250 ml Erlenmeyer flask. Immediately, 1 ml of dibucaine (dissolved in distilled water at 25 mg/ml) was added to the flask and gently swirled for exactly 60 s in an ice water bath. To stop the reaction, 75 ml of ice-cold SPP media (supplemented with 1 mM ethylene glycol-bis(β-aminoethyl ether)-N,N,N′,N′-tetraacetic acid (EGTA)) was immediately added to the Erlenmeyer flask and then split into two 50 ml conical tubes for centrifugation at 2000 × *g* for 10 min at 4°C with no deceleration (Sorvall ST 16R, Rotor 75003181). The supernatant that contained cilia was transferred to centrifuge tubes for the Beckman Coulter JA 25.50 rotor, approximately 30 ml per tube, for centrifugation at 17,000 × *g* for 40 min at 4°C with slow deceleration (Avanti, Rotor JA25.50). The pellet was gently washed with Cilia Wash Buffer (50 mM 4-(2-hydroxyethyl)-1-piperazineethanesulfonic acid (HEPES) at pH 7.4, 3 mM MgSO$_4$, 0.1 mM EGTA, 1 mM DTT, 250 mM sucrose) and frozen with liquid nitrogen for storage in a −80°C freezer.

### Purification of DMT fraction

The cilia suspension was thawed on ice and then centrifuged at 16,000 × *g* and 4°C for 10 min in a microfuge in a refrigerated room (Eppendorf, Centrifuge 5415 D). The pellet was resuspended in 250 μl of ice-cold Cilia Final Buffer (50 mM HEPES at pH 7.4, 3 mM MgSO$_4$, 0.1 mM EGTA, 1 mM DTT, 0.5% trehalose, 1 mM phenylmethylsulfonyl fluoride (PMSF)). To clean the cilia, the resuspended cilia were centrifuged at 16,000 × *g* for 10 min at 4°C in a microfuge (Eppendorf, Centrifuge 5415 D), and the supernatant was removed. Then, the pellet was resuspended in 250 μl of Cilia Final Buffer without trehalose but with 44.1 μl of 10% NP-40 alternative (Millipore Sigma, 492016) added to a final concentration of 1.5% NP-40. The sample was placed on ice to incubate for 30 min before the demembraned flagella supernatant was removed after centrifugation at 16,000 × *g* for 10 min at 4°C in a microfuge

(Eppendorf, Centrifuge 5415 D). The pellet containing the axoneme was resuspended in 245 µl of Cilia Final Buffer (without trehalose), and then 2.5 µl of 100 mM adenosine diphosphate (ADP) (to a final concentration of 1 mM ADP) was added for incubation at room temperature for 10 min. This was followed by adding 2.5 µl of 10 mM adenosine triphosphate (ATP) (to a final concentration of 1 mM) for incubation at room temperature for 10 min. Bradford reagent (Bio-Rad 5000201) was used to measure the total protein concentration, and the final protein concentration was adjusted to 3 mg/ml using Cilia Final Buffer (without trehalose).

## Cryo-EM sample preparation

The concentration of the DMT solution was adjusted to 3 mg/ml. Quantifoil R2/2 grids (Electron Microscopy Sciences, #Q225CR-06) were treated using 1 ml of chloroform overnight followed by negative glow discharge (30 s at 25 mAh). Then, 3.5 µl of axoneme sample was applied to treated grids inside the Vitrobot Mark IV (Thermo Fisher) at a blot force of 3, blot time of 5 s, and drain time of 0.5 s, followed by plunge freezing into liquid ethane.

## Cryo-EM data acquisition

Using a Titan Krios 300 kV FEG electron microscope (Thermo Fisher) with a K3 Summit direct electron detector (Gatan, Inc) and the BioQuantum energy filter (Gatan, Inc), movies of the axoneme were acquired at 64 kx nominal magnification (calculated pixel size of 1.370 Å/pixel) using SerialEM (*Mastronarde, 2005*). A total dose of 45 electrons per $Å^2$ over 40 frames for the *WT* and *MEC17-KO* datasets. A total dose of 73 electrons per $Å^2$ per frame over 30 frames for the *K40R* dataset. The defocus range was between −1.0 and −3.0 µm at an interval of 0.25 µm.

## Cryo-EM image processing

Motion correction and dose-weighting of the movies were performed using MotionCor2 (*Zheng et al., 2017*) implemented in Relion 3 (*Zivanov et al., 2018*), and the contrast transfer function parameters were estimated using Gctf (*Zhang, 2016*). Micrographs with apparent drift, ice contamination, and poor contrast transfer function estimation were discarded (18,384, 25,610, and 4283 micrographs for *WT*, *K40R*, and *MEC17-KO* data, respectively). The filaments were picked manually using e2helixboxe (*Tang et al., 2007*).

An 8 nm periodicity was used to pick particles of 512 × 512 pixels, binned twice, and prealigned using a modified version of the Iterative Helical Real Space Reconstruction script (*Egelman, 2007*) in SPIDER (*Frank et al., 1996*) to work with non-helical symmetry. The alignment parameters were then transferred to Frealign to align the particles for six iterations in Frealign (*Grigorieff, 2007*) and then converted into Relion 3.0. In Relion 3, iterative per-particle-defocus refinement and Bayesian polishing were performed for the 80 nm particles.

Particles were subtracted from their tubulin lattice signal and underwent 3D classification into two classes to obtain the 16 nm repeat particles. The 16 nm repeat particles were then subjected to 3D classification into three classes to obtain the 48 nm repeat particles. The 48 nm particles were then refined, resulting in resolutions of 3.5 and 4.3 Å for *K40R* and *MEC17-KO* DMTs from 182,387 and 39,417 particles, respectively.

To improve the local resolution for each PF during modeling, we performed focused refinements by using masks to cover adjacent PF regions in the DMT. Next, the maps were enhanced by DeepEmhancer (*Sanchez-Garcia et al., 2021*) to improve visualization and interpretability.

## Tubulin modeling

The *WT* tubulin model 6U0H (*Ichikawa et al., 2019*) was first fitted into the higher-resolution *K40R* cryo-EM map and then locally modeled using Coot (*Emsley et al., 2010*) and real space refined in Phenix (*Adams et al., 2010*) for generation of the *K40R* tubulin model. The *WT* and *MEC17-KO* tubulin models were generated by fitting the *K40R* tubulin model in *WT* and *MEC17-KO* cryo-EM maps, respectively, followed by refinement in Coot (*Emsley et al., 2010*) and Phenix (*Adams et al., 2010*). The αK40 loop regions from *WT*, *K40R*, and *MEC17-KO* were locally modeled in focused refinement cryo-EM maps using Coot (*Emsley et al., 2010*) and Phenix (*Adams et al., 2010*). All the maps and model visualization were taken using ChimeraX (*Goddard et al., 2018*).

## Coarse-grained molecular dynamic simulation

Based on the atomic structures of the four tubulin dimers, CFAP52 and IJ34, we performed coarse-grained molecular dynamic simulations. The purpose of this simulation was to check the effect of acetylated αK40 on the binding stability of CFAP52. In the coarse-grained model, each amino acid was represented as a single bead located at its Cα position. To observe their dynamics, we used the excluded volume effect, electrostatic interaction, and the energy function AICG2+ (*Li et al., 2012*; *Li et al., 2014*). In AICG2+, the reference structure was assumed to be the most stable conformation, and their parameters were modified from the reference. It is known that the intradimer interaction is much stronger than the interdimer interaction, and the interdimer interaction is much stronger than the intra-PF interaction, so we set the interdimer and PFs' non-local native interaction force to 0.8 and 0.3 times the original value, respectively, while that of the intradimer was kept as the original value (1.0 times). Of note, three residues (PHE133, GLY308, and GLU401) of the B9-PF β-tubulin at the plus end side were anchored in their position for convenience analysis. We performed the simulation 10 times with acetylated αK40 and non-acetylated αK40 structures using the CafeMol package version 2.1 (*Kenzaki et al., 2011*). Each molecular dynamic simulation took $3 \times 10^7$ molecular dynamic steps, and they were conducted by the underdamped Langevin dynamics at 300 K temperature. We set the friction coefficient to 0.02 (CafeMol unit), and default values were used for others.

Normally, in dealing with electrostatic interactions, LYS and ARG, GLU and ASP, and other amino acids were given a charge of +1, −1, and 0, respectively. However, in this simulation, it was necessary to evaluate the electrostatic interaction as accurately as possible, so we calculated the surface charge density from the all-atom structure and remapped the charge distribution using only Ca beads to reproduce the all-atom surface charge distribution. The technique is called RESPAC (*Terakawa and Takada, 2014*). We applied RESPAC to regions without missing data, such as the αK40 loop and E-hook. For the missing region (and so we performed loop modeling by MODELLER; *Sali and Blundell, 1993*), we treat their charge by default definition. In the coarse-grained model, each amino acid is treated as a single bead, so we simply assumed that if αK40 was acetylated, its charge was zero, while non-acetylated αK40 had a +1 charge.

## All-atom molecular dynamic simulation

The purpose of the all-atom molecular dynamic simulation was to check whether the acetylated αK40 loop takes fewer conformations than the non-acetylated αK40 loop. In the all-atom molecular dynamic simulation using GROMACS (*Abraham et al., 2015*; *Pronk et al., 2013*), we used the GROMOS54a7 force field for protein (*Schmid et al., 2011*) and SPC for solvent water (*Jorgensen and Tirado-Rives, 2005*). We added sodium and chloride ions to neutralize the system and to make the salt concentration approximately equal to 0.1 Energy minimization by the steepest descent minimization algorithm was followed by equilibration with NVT and NPT for 100 ps at 300 K. In the production run, we used NPT ensemble simulations with 1 atm and 300 K. The production run consisted of a 1 fs step for 180 ns. The α-tubulin at the plus end of B9-PF was used as a reference structure for the simulation. To model the structure of acetylated αK40, Vienna-PTM 2.0 was used (*Margreitter et al., 2013*; *Margreitter et al., 2017*; *Petrov et al., 2013*). The force field used (GROMOS54a7) had already been set up for the acetylated lysine.

## Measurement of inter-PF rotation angles

The inter-PF rotation angle can be defined by the lateral rotation angle between each subsequent PF pair. The rotation angles and Z-shift between PF pairs were measured using the 'measure' command from ChimeraX (*Pettersen et al., 2004*). Data for *WT*, *K40R*, and *MEC17-KO* cells were compiled into GraphPad Prism 9 to perform analysis of variance (ANOVA) and plotting.

To perform statistical analysis of the rotation angles, we performed independent cryo-EM reconstructions of two *WT* datasets (*WT* G1 and G2) and two *K40R* datasets (*K40R* G1 and G2) (*Figure 3—figure supplement 1A, B*). These datasets represent biological duplicates with cilia from cells cultured on different dates and prepared for cryo-EM on different dates. Unpaired *t* tests or Mann–Whitney tests were carried out for each angle between both *WT* and *K40R* groups separately. One-way ANOVA or Kruskal–Wallis tests were carried out for each angle between combined *WT* points, combined *K40R* points, and *MEC17-KO* points.

## Measurement of interdimer distance

We docked in the atomic models of the α- and β-tubulins in the maps of *WT*, *K40R*, and *MEC17-KO*. The interdimer distance was measured between the N9 GTP of α-tubulin and that of the next α-tubulin in the same PF using the 'distance' command from Chimera (*Pettersen et al., 2004*).

## Mass spectrometry

Samples prepared for cryo-EM were used for mass spectrometry analysis. Laemmli buffer at 4× (#1610747, Bio-Rad) was added to the microtubule fraction samples in Cilia Final Buffer buffer so that it was 1×, and 25–30 µg protein was loaded on the sodium dodecyl sulfate–polyacrylamide gel electrophoresis gel. Electrophoresis was performed, but the run was terminated before the proteins entered the separation gel. A band containing all proteins in the sample was then cut out from the gel and subjected to in-gel digestion. The obtained peptides (~2 µg) were chromatographically separated on a Dionex Ultimate 3000 UHPLC. First, peptides were loaded onto a Thermo Acclaim Pepmap (Thermo, 75 µm ID × 2 cm with 3 µm C18 beads) precolumn and then onto an Acclaim Pepmap Easyspray (Thermo, 75 µm × 25 cm with 2 µm C18 beads) analytical column and separated with a flow rate of 200 nl/min with a gradient of 2–35% solvent (acetonitrile containing 0.1% formic acid) over 2 hr. Peptides of charge 2+ or higher were recorded using a Thermo Orbitrap Fusion mass spectrometer operating at 120,000 resolution (FWHM in MS1, 15,000 for MS/MS). The data were searched against the *T. thermophila* protein dataset from UniProt (https://www.uniprot.org/).

Mass spectrometry data were analyzed by Scaffold_4.8.4 (Proteome Software Inc). Proteins with mean values of exclusive unique peptide count of 2 or more in the *WT* mass spectrometry results were used for analysis. Raw mass spectrometry data were normalized by the total spectra. ANOVA statistical tests were applied to *MEC17-KO*, *K40R*, and *WT* mass spectrometry results using biological triplicates. Proteins exhibiting a minimum of twofold increase/decrease and a statistical significance threshold ($p < 0.05$) in mutants compared to *WT* were identified as up- or downregulated.

## Acknowledgements

We thank Drs. Kelly Sears, Mike Strauss, Kaustuv Basu, and Jeannie Mui (Facility for Electron Microscopy Research at McGill University) for helping with data collection and maintenance of the electron microscopes. We thank Amy Wong, Lorne Taylor, and Jean-François Trempe (RI-MUHC Proteomics Platform) for their help with mass spectrometry. SK is supported by a JSPS Overseas Research Fellowship. KHB is supported by grants from the Canadian Institutes of Health Research (PJT-156354) and Natural Sciences and Engineering Research Council of Canada (RGPIN-2022-04774). JG is supported by NIH grants R01GM135444 and R01GM139856, respectively.

## Additional information

### Funding

| Funder | Grant reference number | Author |
|---|---|---|
| Natural Sciences and Engineering Research Council of Canada | RGPIN-2022-04774 | Khanh Huy Bui |
| Canadian Institutes of Health Research | PJT-156354 | Khanh Huy Bui |
| National Institutes of Health | R01GM135444 | Jacek Gaertig |
| National Institutes of Health | R01GM139856 | Jacek Gaertig |
| Japan Society for the Promotion of Science | Overseas Research Fellowship | Shintaroh Kubo |

The funders had no role in study design, data collection, and interpretation, or the decision to submit the work for publication.

## Author contributions

Shun Kai Yang, Formal analysis, Investigation, Visualization, Methodology, Writing – original draft, Writing – review and editing; Shintaroh Kubo, Software, Formal analysis, Investigation, Visualization, Methodology, Writing – review and editing; Corbin Steven Black, Katya Peri, Daniel Dai, Melissa Valente-Paterno, Investigation; Thibault Legal, Formal analysis, Visualization; Jacek Gaertig, Resources, Supervision, Funding acquisition, Writing – original draft, Writing – review and editing; Khanh Huy Bui, Conceptualization, Formal analysis, Supervision, Funding acquisition, Investigation, Writing – original draft, Writing – review and editing

## Author ORCIDs

Shun Kai Yang ⓘ https://orcid.org/0000-0002-2363-1441
Shintaroh Kubo ⓘ http://orcid.org/0000-0002-0946-8879
Corbin Steven Black ⓘ http://orcid.org/0000-0003-2777-6434
Katya Peri ⓘ http://orcid.org/0000-0002-7367-7501
Daniel Dai ⓘ http://orcid.org/0000-0002-9973-0446
Khanh Huy Bui ⓘ http://orcid.org/0000-0003-2814-9889

Reviewer #1 (Public review): https://doi.org/10.7554/eLife.92219.3.sa1
Author response https://doi.org/10.7554/eLife.92219.3.sa2

## Additional files

### Supplementary files

• Supplementary file 1. Counts of K40 loops from the wildtype structure, measurement of the tubulin lattices in wildtype and acetylation mutant cells and the proteins downregulated and upregulated in acetylation mutant cells. (**a**) Count of full, partial, and missing αK40 loops based upon the *K40R* cryo-EM density maps, along with possible microtubule inner protein (MIP) interactions for αK40 loops in each protofilament (PF). (**b**) Inter-PF angles between subsequent PF obtained from cryo-EM maps of *WT*, *K40R*, and *MEC17-KO Tetrahymena* cilia. (**c**) Analysis of variance (ANOVA) test p values and significance of the interdimer distances of tubulin subunits within each PF in *WT*, *K40R*, and *MEC17-KO Tetrahymena* cilia. (**d**) Proteins upregulated at least twofold in the *K40R* mutant compared to *WT*. (**e**) Proteins upregulated least twofold in the *MEC17-KO* mutant compared to the *WT*. (**f**) Proteins downregulated at least twofold in the *K40R* mutant compared to the *WT*. (**g**) Proteins downregulated at least twofold in the *MEC17-KO* mutant compared to the *WT*.

• MDAR checklist

### Data availability

The data produced in this study are available in the following databases: Cryo-EM maps of the 48 nm repeat of *MEC17-KO* DMT: EMDB EMD-40436; Model coordinates of the *MEC17-KO* DMT: PDB 8SF7; Mass spectrometry of the cilia from *WT*, *K40R* and *MEC17-KO*, *RIB72A/B-KO*, *RIB72B-KO*: Dryad DOI:10.5061/dryad.3j9kd51sh. The *WT* and *K40R* structures used in this paper have associated maps in the Electron Microscopy Data Bank database with the following EMDB IDs EMD-29685 and EMD-29692.The *WT* and *K40R* structures used in this paper have coordinates in the RCSB Protein Data Bank database with the following PDB IDs 8G2Z and 8G3D.

The following datasets were generated:

| Author(s) | Year | Dataset title | Dataset URL | Database and Identifier |
|---|---|---|---|---|
| Bui KH, Dai D, Yang SK | 2024 | Mass spectrometry of axonemes from *Tetrahymena thermophila* CU428 and acetylation mutants | https://doi.org/10.5061/dryad.3j9kd51sh | Dryad Digital Repository, 10.5061/dryad.3j9kd51sh |

*Continued on next page*

*Continued*

| Author(s) | Year | Dataset title | Dataset URL | Database and Identifier |
|---|---|---|---|---|
| Black CS, Bui KH, Kubo S, Yang SK | 2024 | 48 nm repeat of the doublet microtubule from *Tetrahymena thermophila* strain *MEC17-KO* | https://www.ebi.ac.uk/emdb/EMD-40436 | EMDB, EMD-40436 |
| Black CS, Bui KH, Kubo S, Yang SK | 2024 | 48 nm repeat of the doublet microtubule from *Tetrahymena thermophila* strain *MEC17-KO* | https://www.rcsb.org/structure/8sf7 | RCSB Protein Data Bank, 8SF7 |

The following previously published datasets were used:

| Author(s) | Year | Dataset title | Dataset URL | Database and Identifier |
|---|---|---|---|---|
| Black CS, Kubo S, Yang SK, Bui KH | 2023 | 48-nm doublet microtubule from *Tetrahymena thermophila* strain CU428 | https://www.ebi.ac.uk/emdb/EMD-29685 | Electron Microscopy Data Bank, EMD-29685 |
| Black CS, Kubo S, Yang SK, Bui KH | 2023 | 48-nm doublet microtubule from *Tetrahymena thermophila* strain K40R | https://www.ebi.ac.uk/emdb/EMD-29692 | Electron Microscopy Data Bank, EMD-29692 |
| Black CS, Kubo S, Yang SK, Bui KH | 2023 | 48-nm doublet microtubule from *Tetrahymena thermophila* strain CU428 | https://www.rcsb.org/structure/8G2Z | RCSB Protein Data Bank, 8G2Z |
| Black CS, Kubo S, Yang SK, Bui KH | 2023 | 48-nm doublet microtubule from *Tetrahymena thermophila* strain K40R | https://www.rcsb.org/structure/8G3D | RCSB Protein Data Bank, 8G3D |

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
