## [Editor Report · eLife assessment]

This **fundamental** study employs a combination of cryo-electron microscopy, molecular dynamics, and mass spectrometry to elucidate the role of α-tubulin acetylation at the lumenal lysine 40 residue (αK40) within the cilium. **Compelling** evidence shows αK40 acetylation to impact the structure and stability of doublet microtubules in cilia by affecting the lateral rotational angle. The work will be of relevance to those interested in cytoskeleton and structural biology.

---

## [Referee Report · Reviewer #1 (Public review)]

Summary:

The study "Effect of alpha-tubulin acetylation on the doublet microtubule structure" by S. Yang et al employs a multi-disciplinary approach, including cryo-electron microscopy (cryo-EM), molecular dynamics, and mass spectrometry, to investigate the impact of α-tubulin acetylation at the lysine 40 residue (αK40) on the structure and stability of doublet microtubules in cilia. The work reveals that αK40 acetylation exerts a small-scale, but significant, effect by influencing the lateral rotational angle of the microtubules, thereby affecting their stability. Additionally, the study provided an explanation of the relationship between αK40 acetylation and phosphorylation within cilia, despite that the details still remain elusive. Overall, these findings contribute to our understanding of how post-translational modifications can influence the structure, composition, stability, and functional properties of important cellular components like cilia.

Strengths:

(1) Multi-Disciplinary Approach: The study employs a robust combination of cryo-electron microscopy (cryo-EM), molecular dynamics, and mass spectrometry, providing a comprehensive analysis of the subject matter.

(2) Significant Findings: The paper successfully demonstrates the impact of αK40 acetylation on the lateral rotational angles between protofilaments (inter-PF angles) of doublet microtubules in cilia, thereby affecting their stability. This adds valuable insights into the role of post-translational modifications in cellular components.

(3) Exploration of Acetylation-Phosphorylation Relationship: The study also delves into the relationship between αK40 acetylation and phosphorylation within cilia, contributing to a broader understanding of post-translational modifications.

(4) High-quality data: The authors are cryo-EM experts in the field and the data quality presented in the manuscript is excellent.

(5) Depth of analysis: The authors analyzed the effects of αK40 acetylation in excellent depth which significantly improved our understanding of this system.

Weaknesses:

I have no major concerns about this paper.

---

## [Author Response]

The following is the authors’ response to the original reviews.

We would like to thank the reviewer for the constructive comments. We have revised the papers to address the concerns. In summary, here is what we included in the revised version.

Statistical analysis using biological replicate datasets for WT and K40R doublet microtubule.Addition figures for statistical analysis and MIP decorations in MEC17-KO and K40R.Revised texts and figures to reflect the new changes, cite proper references and fix small errors throughout the text.

**Reviewer #1 (Public Review):**
Summary:The study "Effect of alpha-tubulin acetylation on the doublet microtubule structure" by S. Yang et al employs a multi-disciplinary approach, including cryo-electron microscopy (cryo-EM), molecular dynamics, and mass spectrometry, to investigate the impact of α-tubulin acetylation at the lysine 40 residue (αK40) on the structure and stability of doublet microtubules in cilia. The work reveals that αK40 acetylation exerts a small-scale, but significant, effect by influencing the lateral rotational angle of the microtubules, thereby affecting their stability. Additionally, the study provided an explanation of the relationship between αK40 acetylation and phosphorylation within cilia, despite that the details still remain elusive. Overall, these findings contribute to our understanding of how post-translational modifications can influence the structure, composition, stability, and functional properties of important cellular components like cilia.Strengths:(1) Multi-Disciplinary Approach: The study employs a robust combination of cryo-electron microscopy (cryo-EM), molecular dynamics, and mass spectrometry, providing a comprehensive analysis of the subject matter.(2) Significant Findings: The paper successfully demonstrates the impact of αK40 acetylation on the lateral rotational angles between protofilaments (inter-PF angles) of doublet microtubules in cilia, thereby affecting their stability. This adds valuable insights into the role of post-translational modifications in cellular components.(3) Exploration of Acetylation-Phosphorylation Relationship: The study also delves into the relationship between αK40 acetylation and phosphorylation within cilia, contributing to a broader understanding of post-translational modifications.(4) High-quality data: The authors are cryo-EM experts in the field and the data quality presented in the manuscript is excellent.(5) Depth of analysis: The authors analyzed the effects of αK40 acetylation in excellent depth which significantly improved our understanding of this system.

Thank you for highlighting the strength of our paper.

Weaknesses:I have no major concerns about this paper, but would recommend that a few minor issues be addressed.(1) Lack of Statistical Details: The review points out that the paper could benefit from providing more statistical details, such as the number of particles and maps used for analysis, randomization methods, and dataset splitting for statistical analyses.

To address this, we analyzed the true biological replicate datasets (different cultures, cryo-EM vitrification and data collection) from WT and K40R. Since the MEC17-KO was collected as only one dataset, we decided to not divide the MEC-17 using randomization since the division does not lead to independent sets, which tends to yield identical results in the case of cryo-EM. The biological replicates help us to see how consistent is our structure data for interpretation. The information about the replicate dataset is now included in Table 1. The description of the analysis is highlighted in the manuscript and included in the Materials & Methods and Fig. S4.

In summary, the biological replicate between the WT data indicates that the inter-PF rotation angles are significantly consistent between two biological replicates. On the other hand, there are variations in the inter-PF angles between two replicates of K40R data in the B-tubule (Fig. S4B).

Overall, when pooling the data together ( 6 + 6 measurement points for WT dataset 1 & 2 and 6 + 6 measurement points for K40R dataset 1 & 2 and 6 measurement points for MEC17-KO) (Fig. S4), our analysis yields the same statistical significance as the average of all datasets (6 measurement points of the total averages for WT, K40R and MEC17-KO) (Fig. 3).

In addition, the variation in inter-PF rotation angles between certain PF pairs within the K40R replicates (B7B8 and B9B10) is similar to the variation to MEC17-KO. This suggests that the deacetylation induces variation in inter-PF angles while the inter-PF angles are maintained consistently in WT.

(2) Questionable Conclusion Regarding MIPs: The reviewer suggests caution in the paper's conclusion that "Acetylation of αK40 does not affect tubulin and MIPs." The reviewer recommends that this conclusion be more specific or supported by additional evidence to exclude all other possibilities.

We now revised the text to make sure we do not overclaim that “Acetylation of αK40 does not affect tubulin and MIPs.” We now describe more specifically as “Lack acetylation of αK40 does not significantly affect tubulin and MIP interactions”. Also the text was edited to make the statement more specific.

(3) Need for Additional Visual Data: The reviewer recommends that an enlarged local density map along with fitted PDB models be provided in a supplementary figure, such as Figure 4.

We now include the density maps and fitted PDB models in Fig. 4 and Fig. S5. We also include more snapshots of the MIP in K40R and MEC17-KO in Figure S3.

Overall, the paper is strong in its scientific approach and findings but could benefit from additional statistical rigor and clarification of certain conclusions.Page 11, Line 226: "cluster consists of only ~ acetylated", lacks the percentage. Please correct this.

We corrected it.